# Genome-wide functional screens enable the prediction of high activity CRISPR-Cas9 and -Cas12a guides in *Yarrowia lipolytica*

Dipankar Baisya[1,6], Adithya Ramesh[2,6], Cory Schwartz[2,5], Stefano Lonardi [1,3 ✉] & Ian Wheeldon [2,3,4 ✉]

Genome-wide functional genetic screens have been successful in discovering genotype-phenotype relationships and in engineering new phenotypes. While broadly applied in mammalian cell lines and in *E. coli*, use in non-conventional microorganisms has been limited, in part, due to the inability to accurately design high activity CRISPR guides in such species. Here, we develop an experimental-computational approach to sgRNA design that is specific to an organism of choice, in this case the oleaginous yeast *Yarrowia lipolytica*. A negative selection screen in the absence of non-homologous end-joining, the dominant DNA repair mechanism, was used to generate single guide RNA (sgRNA) activity profiles for both SpCas9 and LbCas12a. This genome-wide data served as input to a deep learning algorithm, DeepGuide, that is able to accurately predict guide activity. DeepGuide uses unsupervised learning to obtain a compressed representation of the genome, followed by supervised learning to map sgRNA sequence, genomic context, and epigenetic features with guide activity. Experimental validation, both genome-wide and with a subset of selected genes, confirms DeepGuide's ability to accurately predict high activity sgRNAs. DeepGuide provides an organism specific predictor of CRISPR guide activity that with retraining could be applied to other fungal species, prokaryotes, and other non-conventional organisms.

[1] Department of Computer Science and Engineering, University of California, Riverside, CA 92521, USA. [2] Department of Chemical and Environmental Engineering, University of California, Riverside, CA 92521, USA. [3] Integrative Institute for Genome Biology, University of California, Riverside, CA 92521, USA. [4] Center for Industrial Biotechnology, University of California, Riverside, CA 92521, USA. [5] Present address: iBio Inc., San Diego, CA, USA. [6] These authors contributed equally: Dipankar Baisya, Adithya Ramesh. ✉email: stelo@cs.ucr.edu; wheeldon@ucr.edu

Class II CRISPR endonucleases such as Cas9 and Cas12a are now widely used for targeted genome editing and in functional genomics screens. These multi-domain proteins function by forming a ribonucleoprotein complex of a CRISPR RNA (crRNA or spacer) and a structural component that enables complexation of the crRNA with the CRISPR-associated endonuclease (i.e., Cas9 or Cas12a)[1,2]. Targeting is achieved by the complementarity of the crRNA to a desired genomic locus, which must be adjacent to a protospacer adjacent motif (PAM) to activate endonuclease function. When this targeting occurs, active Cas9 or Cas12a can create a loss of function mutation as an endonuclease induced double-stranded break in the genome is repaired by native non-homologous end joining (NHEJ) or by homologous recombination (HR) in the presence of a repair template[3,4]. Gene regulation is also possible with Cas activity disabled, by targeting repressor or activation domains to the promoter region of the gene of interest[5]. Such editing and regulation can be accomplished individually[6], in multiplexed format[7] or with pooled libraries of gRNAs that target every gene in a genome[8]. The development of these systems has not only enabled genetic studies in model cell lines and microbes but has also eased the burden of developing targeted genome editing tools in many non-model or non-conventional organisms[9–14].

The successful application of CRISPR systems is largely dependent on the efficacy of the sgRNA, and while a number of design tools have been developed, accurate predictions across species and across different Cas endonucleases are not yet possible. A central challenge is that the vast majority of predictive algorithms are trained on data generated from a limited number of species, most commonly human and murine cell lines or *Escherichia coli*. In addition, most screens to date that correlate sgRNA sequence with activity have been conducted with Cas9 or Cas9 variants, with only a limited number of such screens for Cas12a (Cpf1) or other Cas proteins. A recent meta-analysis of CRISPR-Cas9 screens suggests that the lack of cross-species predictive power comes from variation in genomic context; a strong correlation between sgRNA features and guide activity for the target species was not able to predict guide activity when applied to other species[15]. We have also observed this in our own work, where genome-wide sgRNA activity profiles in the oleaginous yeast *Yarrowia lipolytica* showed poor correlation with activity predicted by a number of commonly-used guide design tools trained on data generated from other species[8].

Here, we developed a deep learning-based guide design algorithm called DeepGuide that is capable of accurately predicting *Streptococcus pyogenes* Cas9 and *Lachnospiraceae bacterium* Cas12a sgRNA activity in *Y. lipolytica*. We focused our efforts on this non-conventional yeast because it has value as an industrial host for the conversion of biomass-derived sugars and industrial waste streams (e.g., glycerol, alkanes, and fatty acids) into value-added chemicals and fuels[16–21]. Similar to many other eukaryotes, DNA repair in *Yarrowia* is dominated by NHEJ[22]. We exploit this trait to perform negative selection CRISPR screens in the absence of NHEJ repair where double-stranded breaks in the genome lead to cell death or a significant impairment to cell fitness[8,23]. Such screens enable the quantification of a cutting score (CS), a measure of activity, for every plasmid expressed sgRNA in the library, thus creating a large dataset correlating sgRNA activity to guide sequence, genomic context, and other genomic and epigenetic features. This work generates a dataset for Cas12a and also uses Cas9 genome-wide CS profiles generated in a previous work[8] to create a large, *Y. lipolytica* specific training set to understand and predict guide activity for CRISPR studies in this yeast.

DeepGuide utilizes a deep learning framework based on a convolutional neural network (CNN), which builds on existing sgRNA activity prediction tools such as DeepCRISPR[24] and Seq-deepCpf1[25]. Unsupervised learning was achieved using a convolutional autoencoder (CAE) in a pretraining step to learn the representation of the sgRNA landscape within the genomic context of *Y. lipolytica*. This was followed by supervised learning on a CNN using sequence and a CS value for each sgRNA sequence within the Cas9 and Cas12a datasets, and related chromatin accessibility information for the target site of each sgRNA. Lastly, the predictions of the model were cross-validated to obtain correlations between observed and predicted CS values. The activity of predicted guides was also independently validated by targeting a set of genes whose null mutants generated easily screenable phenotypes. DeepGuide outperformed existing guide activity prediction tools on the *Y. lipolytica* datasets and predicted 20 nt Cas9 sgRNA with an NGG PAM, as well as 25 nt Cas12a sgRNA with a TTTV PAM, with high accuracy.

## Results

**Library design and generating genome-wide CS profiles**. To generate *Y. lipolytica* CS profiles for CRISPR-Cas9 and CRISPR-Cas12a, we designed plasmid-based sgRNA libraries with sixfold and eightfold redundancy for every protein-coding gene in the *Y. lipolytica* genome. The Cas9 library targeted 7854 out of 7919 protein-coding genes annotated in the CLIB89 strain (parent strain of PO1f) of *Y. lipolytica*[26], while the more restrictive PAM sequence of Cas12a (TTTV for Cas12a vs. NGG for Cas9) resulted in a library targeting only 7801 protein-coding genes. Gene coverage of the library as well as distributions of the guides within each library after plasmid construction are shown in Supplementary Fig. 1. Libraries were designed using two distinct approaches: a strategy biased towards active guides for Cas9, and an unbiased strategy for Cas12a. For the Cas9 library, we used the first iteration of sgRNA Designer[27] to rank all possible Cas9 guides in *Y. lipolytica* and selected the top six scoring guides for every targeted gene (Note: experimental analysis of this library was previously accomplished, including CS profiling, and negative and positive selection screens[8]. Here we re-analyze this data and use it as training and validation sets for DeepGuide). For the Cas12a library, sgRNAs were selected at random starting from the 5' end of each gene. With the exception of ensuring that the sgRNAs would have minimal or no off-target effects, no additional criteria were used to design the library. We used only minimal design criteria so that a significant portion of the library would contain poorly active or inactive guides. This unbiased Cas12a library was expected to provide a more informative training set for DeepGuide due to the presence of a higher proportion of "negative" training examples.

The workflow to generate the CS profiles along with the distributions for both Cas9 and Cas12a are shown in Fig. 1, with replicate correlations shown in Supplementary Fig. 2 and Supplementary Table 1. The CS value for each guide is defined as the $\log_2$ ratio of normalized sgRNA abundance in a NHEJ-deficient strain, to that in a strain both deficient in NHEJ and expressing Cas9/12a (Supplementary Data 1 and 2). The lack of Cas activity removes pressure for selection and therefore sgRNA abundance in the control strain was expected to remain relatively constant over the course of the growth screen. Cas9/12a induced double-stranded breaks in a strain deficient in NHEJ causes cell death or significantly impairs growth, thus linking sgRNA abundance (as measured by next-generation sequencing of the recovered sgRNA expression plasmids) to Cas9/12a activity, where high positive CS values indicate high activity guides and negative CS values indicated inactive or poorly active guides.

With CS profiles for both Cas9 and Cas12a in hand, we set out to determine if a number of commonly used guide prediction methods could capture our experimentally determined CS

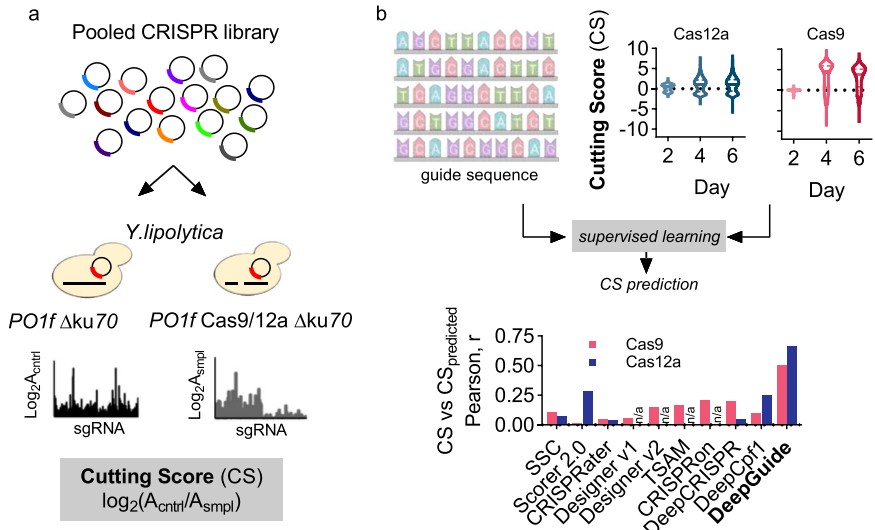

**Fig. 1 Generating genome-wide CRISPR-Cas9 and CRISPR-Cas12a guide activity scores as input to machine learning algorithms for guide activity prediction. a** Pooled libraries of single-guide RNAs (sgRNAs) for *Streptococcus pyogenes* Cas9 and for *Lachnospiraceae bacterium* Cas12a were transformed into *Y. lipolytica* strains with non-homologous end-joining (NHEJ) DNA repair disabled by disruption of KU70. The sample strain (smpl) expresses Cas9 or Cas12a, while the control strain (cntrl) does not. The Cas12a screens were conducted for this work, while the Cas9 screens were previously reported in ref. [8]. A double-stranded CRISPR cut to the genome in the absence of KU70 function leads to cell death (or a dramatic reduction in cell growth), thus enabling the quantification of guide activity through a cutting score (CS) defined as the $\log_2$ fold change of normalized guide abundance in the control vs. the sample determined by next-generation sequencing. **b** Genome-wide CS and sgRNA sequence are used as inputs to the convolutional autoencoder (CAE)-based learning method, DeepGuide, to predict sgRNA CS. DeepGuide prediction of Cas9 guides also used as input a normalized score for nucleosome occupancy across the genome[46]. The performance of established CRISPR guide prediction algorithms, including Spacer Scoring for CRISPR (SSC)[29], sgRNA Scorer 2.0 (Scorer 2.0)[30], CRISPRater[28], Designer v1 and v2[27, 31], TSAM[32], CRISPRon[33], DeepCRISPR[24], and Seq-deepCpf1[25], are shown as a comparison to DeepGuide. The graph shows the Pearson correlation coefficient between CS and the predicted CS for each method. DeepGuide was trained on Cas9 and Cas12a genome-wide CS, the corresponding sgRNA sequence, and genomic context, while all other algorithms used sgRNA sequence (and when appropriate, genomic context) as inputs.

profiles. Learning-based models that use only the sgRNA sequence as input, including CRISPRater[28], SSC[29], and sgRNA Scorer[30] were partially able to capture CS across the genome with SSC exhibiting the highest Pearson coefficient for Cas9 ($r = 0.11$) and sgRNA Scorer the highest for Cas12a ($r = 0.28$). sgRNA Designer[27,31] and TSAM[32] take as input the guide sequence and the genomic context immediately surrounding it but were not able to accurately capture experimentally determined CS values in *Y. lipolytica*. TSAM performed the best of these (including both versions of sgRNA Designer[27,31]), achieving a Pearson coefficient of $r = 0.16$ for Cas9. These three algorithms are not designed for Cas12a guide prediction, as such were not able to predict Cas12a CS in *Y. lipolytica*. Lastly, three neural network-based approaches, Seq-deepCpf1[25], DeepCRISPR[24], and CRISPRon[33], were also only partially aligned with CS; Seq-deepCpf1 fared the best at predicting Cas12a CS ($r = 0.25$), while CRISPRon was best at predicting Cas9 activity ($r = 0.21$). DeepGuide, our CAE/CNN-based approach, achieved Pearson coefficients of 0.5 and 0.66 for Cas9 and Cas12a CS values, respectively. We note here that in the case of Cas9, nucleosome occupancy was also used as input to the predictive algorithm; details of this and DeepGuide optimization are discussed in the following subsections.

The comparison of existing methods to DeepGuide was accomplished using CS values after 4 days of cell growth. CS distributions determined after 2, 4, and 6 days are shown in Fig. 2. After only 2 days of culture, CS values remained close to zero indicating minimal guide activity (at day 2, $CS_{Cas9,avg} = -0.01 \pm 0.21$, $CS_{Cas12a,avg}$ $0.22 \pm 0.83$). At the end of the second day of growth post-transformation, the sample and control strains reached confluency for the first time and were subcultured to continue the growth screen at this time point as well as after reaching confluency for a second time 4 days into the screen. We elected to use day 4 data for further

analysis because the observed CS profiles remained relatively unchanged from day 4 to day 6, suggesting that the majority of sgRNA activity and the resulting phenotypic effect had occurred by day 4. Both libraries also included a population of non-targeting sgRNAs, constituting ~1.5% of each library, that functioned as negative controls. For both Cas12a and Cas9, the average CS for the negative control populations were in the −1.0 to −3.0 range (across all days) and were represented by normal distributions around −1.56 for Cas12a (day 4) and −3.09 for Cas9 (day 4).

**DeepGuide architecture and training.** DeepGuide consists of three interconnected neural networks, namely a CAE, a convolutional FCCN, and a small fully connected network that is used to capture additional epigenetic features (in our case, nucleosome occupancy data; Fig. 3). The CAE takes as input all the *k*-mers from the genome of interest and builds a compressed representation (in the form of internal weights in the encoder) of the genomic background distribution. The second network is composed of an encoder followed by a fully connected neural network (FCCN; see Supplementary Table 2 for the list of layers). The encoder matches the structure of the encoder in the CAE, and its weights are first initialized from the CAE pre-training step. The FCCN is composed of one flattening layer, three fully connected layers, one concatenation layer, and one output layer (see Supplementary Table 3 for the list of layers). The entire second network (including the encoder) is trained via back-propagation from input pairs of sgRNA sequences and their corresponding CS values. The nucleosome data is fed into the third FCCN. One-dimensional occupancy data is expanded into a multi-dimensional real vector using a fully connected layer. The output layer of this third network is finally combined using element-wise multiplication with the output layer of the second

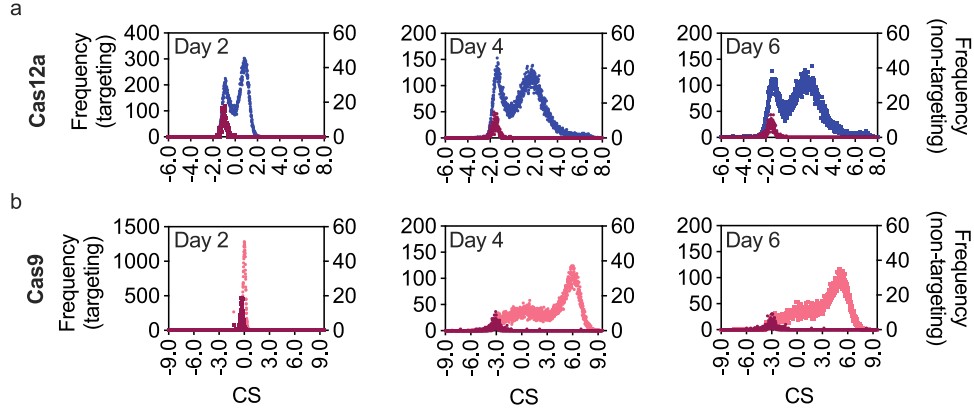

**Fig. 2 CRISPR-Cas12a and CRISPR-Cas9 cutting score (CS) distributions in *Yarrowia lipolytica*.** CS distributions were calculated across 3 separate days after subculturing transformants twice when they reached confluency. Blue and pink distributions plotted on the left *y*-axis show CS values of Cas12a and Cas9 libraries, while the dark red data plotted with the right *y*-axis depicts the non-cutting control population, constituting ~1% of the respective library. The higher the value of CS, the better the cutting activity of the sgRNA. **a** Histogram of CS values in Cas12a library. **b** Histogram of CS values in Cas9 library. The CS values at Day 4 for both Cas9 and Cas12a were carried forward for further analysis.

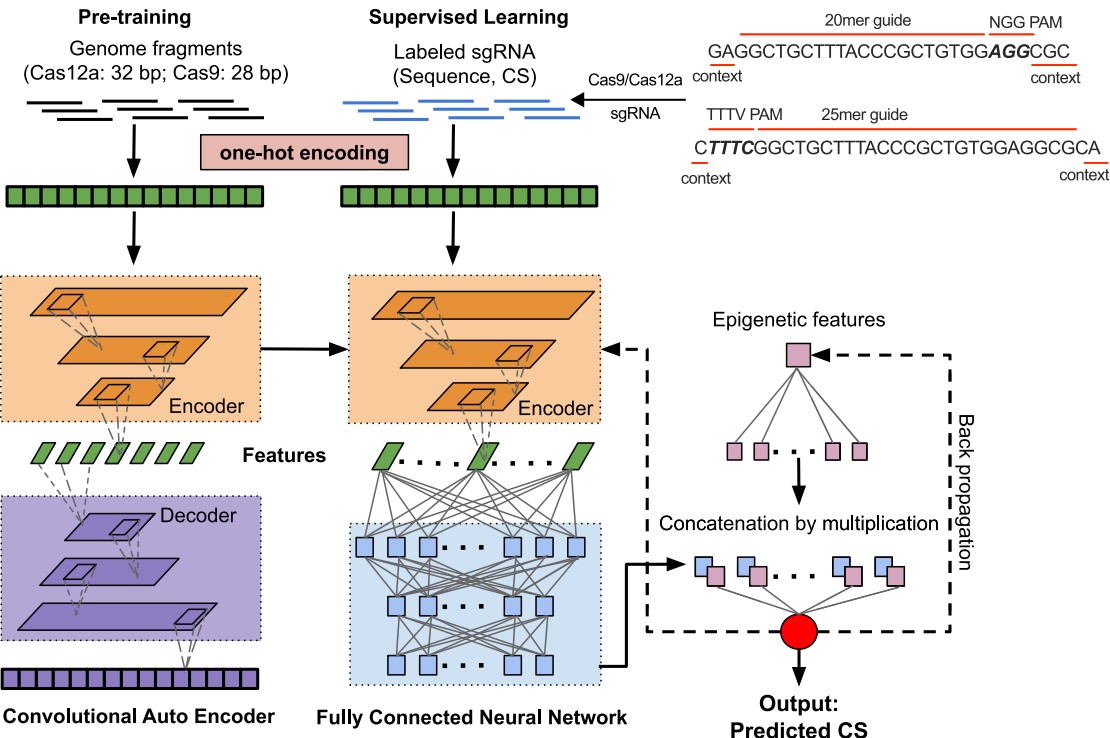

**Fig. 3 The architecture of DeepGuide.** First, the entire *Y. lipolytica* PO1f genome was fragmented into sgRNA sized chunks (using a sliding window of 20 bp for Cas9 and 25 bp for Cas12a). Unsupervised pre-training was carried out on these unlabeled fragments using a convolutional autoencoder (left). The internal weights from the autoencoder were used to initialize a fully connected convolutional neural network (center). Labeled sgRNA (i.e., sequence and associated cutting score) were used as inputs for back-propagation learning on the fully connected neural network. See Supplementary Tables 2 and 3 for a description of the layers.

network to generate CS predictions that account for the sgRNA sequence, genomic context, and nucleosome occupancy. Additional details with respect to these architectures and their training are provided in the "Methods" section.

**DeepGuide optimization.** The choice of a CAE combined with a fully connected CNN was motivated by the results of a five-fold cross-validation performance evaluation among various machine learning methods (Fig. 4a). The compared methods include support

vector machines, gradient boosting, logistic and linear regression, random forests, and a FCNN. As judged by Pearson and Spearman correlations of the predicted CS and experimentally determined CS, the core CAE/CNN architecture of DeepGuide performed better than all other tested methods. For Cas12a, DeepGuide achieved a Pearson *r*-value of 0.66 and a Spearman *r*-value of 0.66, while for Cas9 Pearson and Spearman values were 0.43 and 0.37, respectively. The inclusion of nucleosome occupancy data improved Cas9 prediction accuracy, increasing the Pearson and Spearman *r*-values to 0.50 and 0.43, respectively. This effect is in agreement with

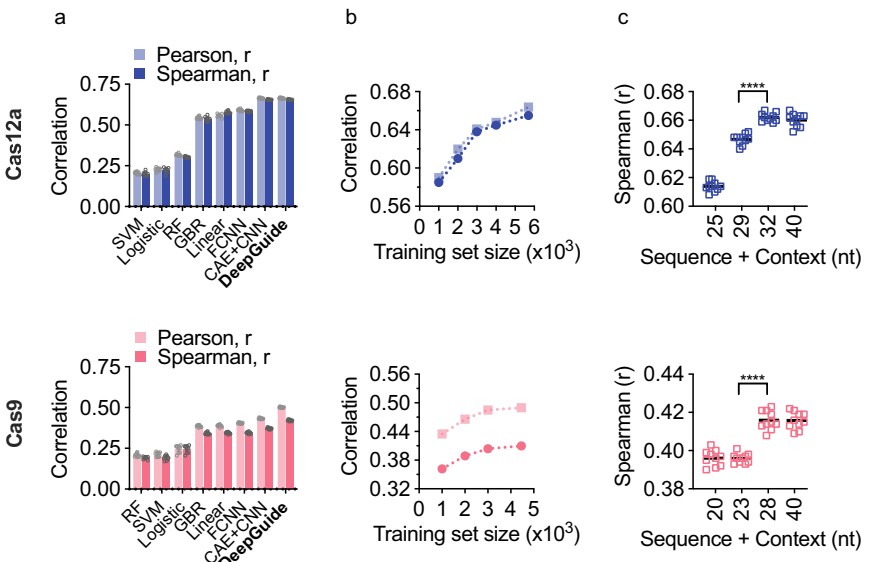

**Fig. 4 Design and parameter optimization for DeepGuide on the Cas12a (top) and Cas9 (bottom) datasets. a** Evaluation of DeepGuide in a cross-validation analysis with several machine learning (ML) methods, including random forest (RF), support vector machines (SVM), logistic regression (Logistic), gradient boosting regression (GBR), linear regression (Linear), fully connected neural networks (FCNN), and the core architecture of DeepGuide, a combination of a convolutional autoencoder and a convolutional fully connected neural network (CAE + CNN). In addition to interconnected CAE and CNN, the final architecture of DeepGuide also includes a third fully connected network to account for nucleosome occupancy. Error bars indicate standard deviation over five independent cross-validation experiments. **b** The dependency of DeepGuide's performance as a function of the training set size with smaller datasets produced by downsampling. **c** The dependency of DeepGuide's performance as a function on the length of the context sequence around the sgRNA (tenfold cross-validation). One-way ANOVA indicates that sequence length has a significant effect (****$p < 0.0001$) for both Cas12a and Cas9. Tukey's multiple comparison post hoc analysis indicates that for Cas12a the Spearman values for all sequence lengths, with exception of 32 vs. 40 bp ($p = 0.708$), are significantly different ($p < 0.0001$). For Cas9, Tukey's multiple comparisons indicates that all values are significantly different ($p < 0.0001$) with the exceptions of 20 vs. 23 ($p = 0.9995$) and 28 vs. 40 bp ($p > 0.9999$).

observations of nucleosome inhibition of Cas9/12a targeting in vitro and in vivo[34–37]. A similar nucleosome occupancy effect on DeepGuide's ability to predict Cas12a CS values, however, was not observed.

One important question about the performance of any machine learning method relates to the size of the training set, that is, how much data are necessary to obtain the best predictions and what performance penalty is incurred when the training dataset size is limited. Figure 4b shows the Pearson and Spearman correlations for DeepGuide as the size of the dataset increases, up to the full-size dataset correlating sgRNA sequence to experimentally determined CS. This analysis shows that (i) DeepGuide's performance improves as the size of the training set increases for both Cas12a and Cas9, and (ii) the performance for Cas9 plateaus as dataset size increases above ~30,000 examples. While the performance curve for Cas12a appears to indicate that a larger dataset could potentially improve performance, the trend still shows that the correlations start to the plateau above a training set size of ~30,000.

DeepGuide's hyperparameters (e.g., number of hidden layers, number of neurons in each layer, type of activation function, learning rate, etc.) were also optimized using cross-validation.

To determine the optimal number of hidden layers in the FCCN downstream of the encoder, we carried out an ablation analysis as described in the next section. Among the input hyperparameters, the length of the context around the sgRNA significantly affected prediction performance. Observe that sequence lengths from 32-40 bp resulted in the best performance for Cas12a; 32 bp was selected because it produced a model with a smaller number of parameters, thus reducing the possibility of overfitting (Fig. 4c). Similarly, for Cas9 28 bp was selected from a range of 20–40 bp as it produced the best prediction performance.

**Ablation analysis of DeepGuide.** To understand how pre-training and the number of fully connected layers (downstream of the encoder in the second network) affect DeepGuide's performance, an ablation analysis was performed. First, as a "sanity" check, the encoder alone (i.e., no fully connected layers, but a flatten layer to get a single output) was tested on Cas12a and Cas9 data without any training or pre-training (i.e., using random weights). Observe in the first row of Table 1 (also see Tables S4 and S5) that Spearman and Pearson are essentially zero, as expected. Second, random weights were used for the encoder, then back-propagation was run on the flatten layer. Observe in the second row that training just one layer resulted in a significant jump in prediction performance on both datasets. In rows 3–7, the weights of the encoder were initialized from the pre-training step (CAE) and back-propagation was run exclusively on the fully connected layers downstream of the encoder, that is by freezing the pre-trained weights of the encoder. Under these conditions, the performance was measured by incrementally adding one fully connected layer at the time. By comparing row 2 to row 3, observe that pre-training improves the performance for both Cas12a and Cas9, but more so for Cas12a. Also, observe in rows 3–7 that the best performance on the Cas12a dataset is obtained when the second network includes only one fully connected layer ($fc_8$). Similarly, rows 3–7 show that none of the fully connected layers ($fc_8$, $fc_9$, $fc_{10}$) help to improve the performance of the Cas9 dataset. However, a significant performance improvement was gained for Cas9 by introducing the multiplication layer ($mult_{11}$), which combines the nucleosome occupancy.

If back-propagation is allowed to fine-tune the weights of the encoder, the overall performance improvement is striking (i.e., compare rows 3–7 with rows 8–12). Observe that in the case of Cas12a, one additional fully connected layer ($fc_9$) helps the

**Table 1 DeepGuide ablation analysis.**

| Row | Training | Layer | Pearson, $r$ | |
|---|---|---|---|---|
| | | | **Cas12a** | **Cas9** |
| 1 | Random weights | Encoder⇨flatten$_7$ | 0.070 | 0.003 |
| 2 | Back prop-flatten$_7$ | Encoder⇨flatten$_7$ | 0.455 | 0.312 |
| 3 | Pretrained + back prop flatten$_7$⇨fc$_{8-10}$⇨mult$_{11}$ | Encoder⇨flatten$_7$ | 0.532 | 0.353 |
| 4 | | Encoder⇨flatten$_7$⇨fc$_8$ | **0.534** | 0.310 |
| 5 | | Encoder⇨flatten$_7$⇨fc$_8$⇨fc$_9$ | 0.517 | 0.291 |
| 6 | | Encoder⇨flatten$_7$⇨fc$_8$⇨fc$_9$⇨fc$_{10}$ | 0.514 | 0.305 |
| 7 | | Encoder⇨flatten$_7$⇨fc$_8$⇨fc$_9$⇨fc$_{10}$⇨mult$_{11}$ | 0.514 | **0.388** |
| 8 | Pretrained + back prop-all | Encoder⇨flatten$_7$ | 0.641 | 0.409 |
| 9 | | Encoder⇨flatten$_7$⇨fc$_8$ | 0.658 | 0.424 |
| 10 | | Encoder⇨flatten$_7$⇨fc$_8$⇨fc$_9$ | **0.664** | 0.414 |
| 11 | | Encoder⇨flatten$_7$⇨fc$_8$⇨fc$_9$⇨fc$_{10}$ | 0.664 | 0.414 |
| 12 | | Encoder⇨flatten$_7$⇨fc$_8$⇨fc$_9$⇨fc$_{10}$⇨mult$_{11}$ | 0.664 | **0.501** |

Row 1 shows the performance of the encoder (followed by a flatten layer) using random weights (no pre-training or back-propagation); row 2 shows the performance of the encoder (followed by a flatten layer) using random weights and then performing back-propagation only on the flatten layer; rows 3–7 show the performance after pre-training the encoder and then running back-propagation only layers downstream of the encoder; rows 8–12 show the performance after pre-training and then running back-propagation on the whole network (including the encoder); correlation coefficients in bold corresponds to the best performance.
*fc* fully connected layer, *flatten* flatten layer, *mult* multiplication layer (see Supplementary Table 3 for the list of layers).

performance but adding more is detrimental. As a result of this ablation analysis, the third fully connected layer (fc$_{10}$) and the multiplication layer (mult$_{11}$) were removed from DeepGuide's architecture for Cas12a guides.

On Cas9, observe in Table 1 that adding one fully connected layer (fc$_8$) improves the performance, but the biggest improvement is due to the multiplication layer (mult$_{11}$) that incorporates the nucleosome occupancy data. As a result of this ablation analysis, the second and third fully connected layers (fc$_9$ and fc$_{10}$) were removed from DeepGuide's architecture for Cas9 guides.

**External and internal validation of DeepGuide.** Given the optimized DeepGuide architecture, we next set out to measure its ability to predict Cas9 and Cas12a sgRNA activity as measured in single-gene disruption experiments. To do so, we used Deep-Guide to predict five high activity and five poor activity Cas9 and Cas12a sgRNAs for four genes whose disruption can be measured with an easily screenable phenotype (Supplementary Fig. 3). These genes included MFE1, the knockout of which prevents growth on long-chain fatty acids; CAN1, which is involved in resistance to L-canavanine; and MGA1 and RAS2, knockouts of which result in colonies with a smooth appearance due to loss of pseudohyphae formation. Plasmids expressing each of the sgRNAs were individually transformed into *Y. lipolytica* in biological triplicate and screened for the presence or absence of the targeted phenotype. For high activity Cas9 guides, the predicted CS ranged from 4.65 to 5.19, while for Cas12a the CS values of the highest activity guides ranged from 1.09 to 2.08. At the lower end, poor-activity guides ranged from −1.12 to 1.88 for Cas9 and −0.72 to 1.00 for Cas12a. The near overlap of CS values in the low and high predicted activity groups for Cas12a is due to the fact that only 12 TTTV PAM sequences are contained within MGA1, thus providing a limited set to select from. The ten guides that provided the largest range were selected even though two of these had nearly equal predicted CS values (for MGA1, CS$_{predicted}$ = 1.09 was included in the high activity group, while CS$_{predicted}$ = 1.00 was included in the low activity group).

DeepGuide was generally successful in predicting active sgRNAs for both Cas12a and Cas9 but had limited ability to accurately predict low-activity guides for Cas9 (Fig. 5). Seventeen of the twenty Cas12a guides that were predicted to be of high activity, clustered together with a mean disruption efficiency of 77.4% and a CS$_{predicted}$ of 1.67 (Supplementary Fig. 4). Three guides from the high-activity group, CS$_{predicted}$ of 1.91, 1.65, and 1.09, did not cluster well with the others and exhibited disruption efficiencies of 24.6, 19.1, and 4.8%, respectively. Predicting the lower end of the activity scale was also successful for Cas12a where 20 of the 20 guides clustered together with an average disruption efficiency of 12.1% and a CS$_{predicted}$ of 0.16. Predictions for highly active Cas9 guides were also accurate; 18 of 20 sgRNAs clustered together with an average disruption efficiency and CS$_{predicted}$ of 69.8% and 4.91, respectively. However, DeepGuide performed poorly in predicting low or inactive guides for Cas9. Only 4 of 20 sgRNAs in the low-activity group exhibited disruption efficiencies below 25%, another 9 sgRNAs achieved efficiencies between 25% and 50%, while the remaining seven proved to be highly active with disruption efficiencies above 50%. One explanation for the discrepancy in performance between Cas9 and Cas12a is the difference in training sets. The Cas9 library was biased from the outset toward high-activity guides, thus limiting the number of poor activity guides to learn from. Conversely, the Cas12a sgRNA library was not biased from the outset and consequently resulted in a CS dataset that included a high number of both poorly active and highly active guides.

In addition to the external validation where individual sgRNAs were tested for disruption efficiency, we also sought to evaluate DeepGuide's ability to discriminate between active and inactive sgRNAs as measured in the pooled screens; that is, we compared experimentally determined CS vs. predicted CS in a receiver operating characteristic curve (ROC) analysis. To do so, the mean CS values of the high activity clusters for Cas9 and Cas12a were taken as the threshold for binarizing guide activity. Guides with CS > 1.67 for Cas12a and CS > 4.91 for Cas9 were classified as active, and CS values below this threshold were classified as inactive. DeepGuide outperformed all other tools in classifying highly active guides as indicated by an area under the ROC (AUROC) of 0.77 for Cas12a and 0.73 for Cas9 (Fig. 5c, d and Supplementary Fig. 5). It is important to note that when seq-DeepCpf1, a prediction algorithm with similar architecture as DeepGuide, was retrained on the Cas9 and Cas12a CS profile generated in *Yarrowia*, the AUROC curve improved from 0.61 to 0.72 for Cas12a and 0.58 to 0.69 for Cas9, underscoring the importance of the dataset used for training the machine learning model.

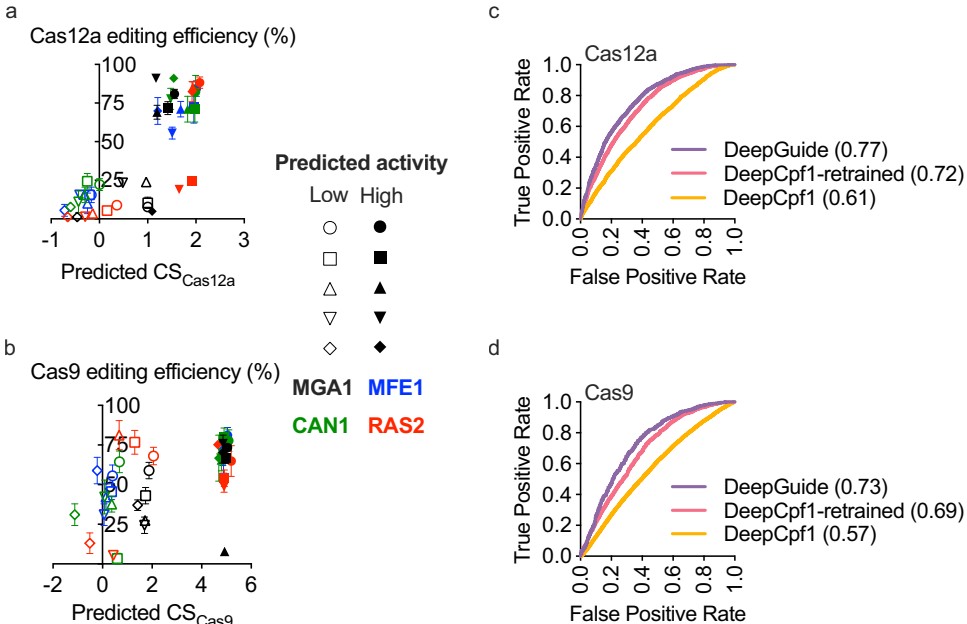

**Fig. 5 External and internal validation of DeepGuide performance. a, b** Editing efficiencies of 5 predicted high-activity and 5 predicted low-activity sgRNA for Cas12a and Cas9 in single-gene disruption experiments. Genes MGA1, MFE1, CAN1, and RAS2 were picked as their null mutants displayed easily screenable phenotypes. Predicted high-activity sgRNAs clustered together, while low-activity sgRNAs clustered at lower editing efficiencies for Cas12a. Data points represent the mean of three biologically independent samples ($n = 3$), while the error bars represent the standard deviation. **c, d** ROC plots and AUROC values for DeepGuide prediction of high- and low-activity Cas9 and Cas12a sgRNAs.

## Discussion

Current prediction methods have proven effective at designing active CRISPR sgRNAs[24,25,27–33,38,39], but the predictive power is typically limited to the organism from which the training data was generated[8,15]. In this context, we created DeepGuide, a machine learning approach to design sgRNA guides based on an organism-specific training set. An evaluation of several machine learning methods and models (see Fig. 1) allowed us to choose the combination of architectures that would achieve the best predictive performance on our *Y. lipolytica* datasets. When trained on genome-wide CS profiles for both Cas12a and Cas9, Deep-Guide accurately designed sgRNA sequences that resulted in high genome editing efficiency (Fig. 5) and outperformed other methods in predicting Cas9 and Cas12a activity across the genome. Ablation analysis revealed that the organism-specific nature of DeepGuide is not solely related to the sgRNA training set but also the genomic context; predictions improved for both Cas9 and Cas12a if DeepGuide's internal weights were initialized via a genome-wide unsupervised learning step on the *Y. lipolytica* genome, rather than being assigned at random (Fig. 3 and Table 1). With retraining, DeepGuide was able to predict guide activity in *E. coli* (see ref. [38]) with good accuracy (see Supplementary Fig. 7). Given the significant differences in genomic context and methods of generating genome-wide activity profiles, we found that Yarrowia-optimized DeepGuide was not able to accurately predict sgRNA activity in mammalian cells to the same level as the bidirectional long short-term memory neural networks (LSTM) methods that were highly-optimized on such datasets (see ref. [39]; Supplementary Fig. 7).

While DeepGuide was successful in designing active guides for both Cas12a and Cas9, our analysis and validation experiments revealed significant differences between the two systems. The first was that DeepGuide performed much better on the Cas12a dataset (Cas12a Pearson, $r = 0.66$ vs. Cas9, $r = 0.50$), possibly due to the fact that the Cas12a library covers a greater fraction of the total Cas12a PAM sites within the genome (there are 809,401 TTTN PAM sites for Cas12a in *Y. lipolytica* and 2,415,425 Cas9 NGG PAM sites). Library design could also be a driving factor; DeepGuide was not able to accurately predict poor activity guides for Cas9, a result that we ascribe to the low number of "negative" examples in the biased library designed for Cas9. Lastly, sequence and genomic context were sufficient to drive accurate predictions for Cas12a, but additional contextual information in the form of nucleosome occupancy was necessary to obtain the maximal predictive power for Cas9. The difference in predictive performance between the two systems highlights the importance of having a "good" training set, in particular for deep learning architectures. A good training set for CRISPR sgRNA prediction should represent high and low activity guides equally, should uniformly sample the entire genome-wide $k$-mer space, should be noise-free (i.e., the guide activity scores should be accurate), and should be sufficiently large (e.g., tens of thousands of data points or more).

While this work focuses on the development of DeepGuide for its specific use in *Y. lipolytica*, the same experimental-computational workflow that involves (i) library design, (ii) generating genome-wide guide activity profiles, (iii) predictor design (learning and optimization), and (iv) external validation, can be readily applied to other fungal species, broadly to prokaryotes, and any other organisms in which genome-wide functional screens can be used to estimate sgRNA activities. Moreover, DeepGuide adds to the growing number of examples in which deep learning is being used to solve complex problems in molecular biology, e.g., the prediction of essential genes[40,41].

## Methods

**DeepGuide architecture.** DeepGuide uses a CAE to derive a reduced-dimensionality representation of the underlying distribution of sgRNA sequences in the whole genome. The autoencoder is composed of an encoder (6 layers) and a decoder (6 layers). The objective of the unsupervised training is to infer the internal weight so the input layer to the encoder is as close as possible to the output layer of the decoder. The CAE encoder has two Conv1D layers of 20 filters and 40 filters, respectively, one MaxPooling1D layer, one AveragePooling1D layer, and two

BatchNormalization layers (see Supplementary Table 2 for the order). A rectified linear activation function (ReLU) is used as activation and the Glorot uniform initializer is used to initialize the convolutional filters. The layer regularizer for the encoder is L2 with a value of 10E−4. The decoder has the same structure as the encoder but uses UpSampling1D instead of MaxPooling, and UpSampling1D instead of AveragePooling1D. The layer regularizer in the decoder is again L2 with a value of 10E−4. The loss function for training is the binary cross-entropy, and Adam is the optimizer with a learning rate of 10E−3. A batch size of 64 and 200 epochs are used for training (no early stopping).

The encoder in the second network has the same structure as the encoder in the CAE (see Supplementary Table 3). The initial configuration of the network downstream of the encoder uses one flatten layer, three fully connected layers (fc8, fc9, fc10) of 80 neurons, 40 neurons, and 40 neurons, respectively. The feature map for layer pool6 is $7 \times 40$, which is 280 dimensional. The feature map for the first fully connected layer (fc8) is $280 \times 80 = 22400$ dimensional. The feature map for the second and third fully connected layers (fc9 and fc10) are $80 \times 40 = 3200$ and $40 \times 40 = 1600$ dimensional, respectively. Layer mult11 is a multiplication layer that combines sequence and nucleosome occupancy features. ReLU is the activation and Glorot uniform initializer is used to initialize the convolutional filters. The second network is trained for 150 epochs using back-propagation; if the value of loss function does not improve for 15 consecutive epochs the training is terminated.

The third fully connected network is used to provide DeepGuide with nucleosome occupancy data. The nucleosome occupancy for each sgRNA is a floating-point value in [0,1]. The third network uses one fully connected layer with 40 units to expand the one-dimensional nucleosome occupancy value to a 40-dimensional vector, to match the dimensionality of the output layer of the second network. Sequence and nucleosome data are merged by performing an element-wise multiplication between the output layer of the second network and the output layer of the third network. When DeepGuide is used in "classification mode" (i.e., binary output) the activation function is a sigmoid; when DeepGuide is used in "regression mode" (i.e., CS output), the activation function is linear.

Note that following the ablation analysis, only two fully connected layers (and no multiplication layer) are used for Cas12a; similarly, only one fully connected layer connected to the multiplication layer is used for Cas9.

**DeepGuide training and pre-training**. For the pre-training step of the CAE all $k$-mers from the *Y. lipolytica* genome were extracted using a sliding window of 1 bp. For Cas9 the input length was 28 bp, which includes the length of each possible spacer (20 bp), plus 3 bp for a PAM sequence, and 2 bp upstream and downstream for context. For Cas12a, 32-mers were used to account for the 25 bp spacer, a 4 bp PAM, 1 bp of context upstream of the PAM, and 2 bp of context downstream of the spacer (see Fig. 4b). These unlabeled sgRNA datasets contained over 20 million $k$-mers each. sgRNA sequences were converted into a numerical representation using one-hot encoding, that is, each sgRNA was converted to a $4 \times n$ dimensional binary matrix where $n$ is the length of the guide.

The training data to DeepGuide consisted of sgRNA sequences, their nucleosome occupancy score, and their CS values. sgRNA sequences were one-hot encoded, while nucleosome occupancy data were processed as explained in the "Nucleosome occupancy analysis" subsection below. CS scores were produced as explained in the "CS analysis" subsection also provided below.

When the pre-training concluded, the internal weights of the CAE were used to initialize the encoder in the second network. The second network was trained via back-propagation using either ~45,000 sgRNAs for Cas9 or ~58,000 sgRNA for Cas12a, each with their associated CS value. In all, 60% of these guides were used for training, 20% for validation, and 20% for testing. The training step not only allowed the inference of the weights for the fully connected layers downstream of the encoder but also fine-tuned the weights of the encoder. As explained in the section "Ablation analysis of DeepGuide" (main text) the pre-training step helped the supervised learning to converge faster and improved the prediction performance.

Supplementary Fig. 6 illustrates the loss curve for training and validation of the CNN without pre-training and with pre-training as a function on the number of training epochs. Observe that in the CNN without pre-training the difference between training and validation loss function starts increasing after about 20 epochs. In contrast, for the CNN with pre-training, the training and validation curves of the loss function are overlapping after about 30 epochs. This indicates that the pre-training prevents the network from overfitting and helps the network to generalize better.

**sgRNA library design**. Custom Matlab scripts were used to design an LbCas12a sgRNA library with ~8-fold coverage of all protein-coding sequences annotated in the *Y. lipolytica* PO1f parent strain genome, CLIB89 [https://www.ncbi.nlm.nih.gov/assembly/GCA_001761485.1][26]. A list of 25 nucleotide (nt) sgRNAs with a TTTV (V = A/G/C) PAM were identified in both the top and the bottom strand of the coding sequence of each gene (CDS). A second list containing all possible 25nt sgRNAs with a TTTN PAM from the top and bottom strands of all 6 chromosomes in *Y. lipolytica* was also generated and used to test for sgRNA uniqueness. The uniqueness test was carried out by comparing the first 14nt of each sgRNA in the first list to the first 14nt of every sgRNA in the second list. If a sequence occurred

more than once, the sgRNA was identified as non-unique and excluded from consideration. The sgRNAs that passed the test for uniqueness were then picked in an unbiased manner, with even representation from the top and bottom strands when possible, starting from the 5' end of the CDS. Six-hundred and fifty-one sgRNAs of random sequence confirmed to not target in the genome were also designed using a similar methodology but with more stringent criteria for uniqueness (i.e., first 10 nt were not found anywhere in the genome). A detailed procedure of sgRNA design for both Cas9 and Cas12a is provided in ref. [42] and additional data on the Cas9 guide design criteria are provided in ref. [8]. Briefly, for Cas9 sgRNAs the first version of sgRNA Designer[27] was used to identify the top predicted guides for every CDS, these guides were filtered for uniqueness, and the top six unique guides were selected.

**Microbial strains and culturing**. The parent yeast strain used in this study was *Y. lipolytica* PO1f with genotype MatA, leu2-270, ura3-302, xpr2-322, axp-2. The PO1f Cas9 and the PO1f Cas12a strains were constructed by integrating UAS1B8-TEF(136)-Cas9-CYCT and UAS1B8-TEF(136)-LbCpf1-CYCT expression cassettes into the A08 locus[43]. The PO1f Cas9 *ku70* and PO1f Cas12a *ku70* strains were constructed by disrupting KU70 using CRISPR-Cas9 as previously described[23]. All strains used in this study are listed in Supplementary Table 6. All plasmid construction and propagation were conducted in *E. coli* TOP10. Cultures were conducted in Luria-Bertani (LB) broth with 100 mg L⁻¹ ampicillin at 37 °C in 14 mL polypropylene tubes, at 225 r.p.m. Plasmids were isolated from *E. coli* cultures using the Zymo Research Plasmid Miniprep Kit.

**Plasmid construction**. All plasmids and primers used in this work are listed in Supplementary Tables 7 and 8. To create the LbCas12a sgRNA expression plasmid (pLbCas12ayl), we first added a second direct repeat sequence at the 5' of the polyT terminator in pCpf1_yl (see ref. [44]). This was done to ensure that library sgRNAs could end in one or more thymine residues without being construed as part of the terminator. To make this change, pCpf1_yl was first linearized by digestion with SpeI. Subsequently, primers ExtraDR-F and ExtraDR-R were annealed and this double-stranded fragment was used to circularize the vector (NEBuilder® HiFi DNA Assembly) For integrating LbCas12a, pHR_A08_LbCas12a was constructed by digesting pHR_A08_hrGFP (Addgene #84615) with BssHII and NheI, and the LbCas12a fragment was inserted using the New England BioLab (NEB) NEBuilder® HiFi DNA Assembly Master Mix. The LbCas12a fragment was amplified along with the necessary overlaps by PCR using Cpf1-Int-F and Cpf1-Int-R primers from pLbCas12ayl. Successful cloning of the entire fragment was confirmed with sequencing primers A08-Seq-F, A08-Seq-R, Tef-Seq-F, Lb1-R, Lb2-F, Lb3-F, Lb4-F, and Lb5-F. To create the Cas12a sgRNA genome-wide library expression plasmid (pLbCas12ayl-GW) the UAS1B8-TEF- LbCas12a-CYC1 fragment was removed from pLbCas12ayl with the use of XmaI and HindIII restriction enzymes. Subsequently, the primers BRIDGE-F and BRIDGE-R were used to circularize the vector, and the M13 forward primer was used to ensure the correct assembly of the construct.

To conduct the validation experiments of predicted CS values by DeepGuide, four genes with easily screenable phenotypes were selected and 10 sgRNAs (five highly active and five with poor activity) targeting each of these genes for Cas9 and Cas12a were selected and cloned for individual disruption experiments. All 40 Cas9 sgRNAs with required overlaps for cloning were purchased from a commercial vendor (IDT-DNA) as single-stranded primers and assembled into pCRISPRyl (Addgene #70007) after linearizing the vector with AvrII, using NEBuilder® HiFi DNA Assembly. In a similar manner, the 40 Cas12a sgRNAs with necessary overlaps were cloned into pLbCas12ayl, after linearizing the vector with SpeI. These primers are also included in Supplementary Table 8.

**sgRNA library cloning**. The LbCas12a library targeting the protein-coding genes in PO1f was ordered as an oligonucleotide pool from Agilent Technologies Inc. and cloned in-house using the Agilent SureVector CRISPR Library Cloning Kit (Part Number G7556A). The backbone vector (pLbCas12ayl-GW) was first linearized by PCR using the primers InversePCR-F and InversePCR-R, DpnI digested, cleaned up using Beckman AMPure XP SPRI beads, and transformed into *E.coli* TOP10 cells to verify minimal contamination from the circularized plasmid. Library oligos were amplified by PCR using the primers OLS-F and OLS-R for 15 cycles as per vendor instructions using Q5 high fidelity polymerase and cleaned up using the AMPure XP beads. The linearized backbone and the amplicons were combined in 4 replicate reactions of sgRNA library cloning that were carried out as per vendor instructions and pooled prior to bead cleanup. Two amplification bottles containing 1 L of LB media and 3 g of library-grade low gelling agarose were prepared, autoclaved, and cooled to 37 °C. Eighteen replicate transformations of the cloned library were conducted using Agilent's ElectroTen-Blue cells (Catalog #200159) via electroporation (0.2 cm cuvette, 2.5 kV, 1 pulse). Cells were recovered and with a 1 hr outgrowth in SOC media at 37 °C (2% tryptone, 0.5% yeast extract, 10 mM NaCl, 2.5 mM KCl, 10 mM MgCl₂, 10 mM MgSO₄, and 20 mM glucose.) The transformed *E. coli* cells were then inoculated into two amplification bottles and grown for 2 days until colonies were visibly suspended in the matrix. Colonies were recovered by centrifugation and subject to a second amplification step by inoculating an 800 mL LB culture. After 4 hr, the cells were collected, and the pooled

plasmid library was isolated using the ZymoPURE II Plasmid Gigaprep Kit (Catalog #D4202) yielding ~2.4 mg of plasmid DNA containing the Cas12a sgRNA library. The library was subject to a NextSeq run to test for fold coverage of individual sgRNA and skew.

**Yeast transformation and screening**. Transformation of *Y. lipolytica* with the sgRNA plasmid library was done using a previously described method with slight modifications[8]. Briefly, 3 mL of YPD was inoculated with a single colony of the strain of interest and grown in a 14 mL tube at 30 °C with shaking at 200 RPM for 22-24 hours (final OD ~30). Cells were pelleted by centrifugation (6,300 g) and washed with 1.2 mL of transformation buffer (0.1 M LiAc, 10 mM Tris (pH=8.0), 1 mM EDTA). To these resuspended cells, 36 μL of ssDNA mix (8 mg/mL Salmon Sperm DNA, 10 mM Tris (pH = 8.0), 1 mM EDTA), 180 μL of β-mercaptoethanol mix (5% β-mercaptoethanol, 95% triacetin), and 8 μg of plasmid library DNA were added, mixed via pipetting, and incubated for 30 min at room temperature. After incubation, 1800 μL of PEG mix (70% w/v PEG (3350 MW)) was added and mixed via pipetting, and the mixture was incubated at room temperature for an additional 30 min. Cells were then heat shocked for 25 min at 37 °C, washed with 25 mL of sterile milliQ $H_2O$, and used to inoculate 50 mL of SD-leu media for screening experiments. Dilutions of the transformation (0.01% and 0.001%) were plated on solid SD-leu media to calculate transformation efficiency. Three biological replicates of each transformation were performed for each condition. Transformation efficiency for each replicate is presented in Supplementary Table 9. Details of the Cas9 library are provided in ref. [8].

Screening experiments were conducted in 50 mL of liquid media in a 250 mL baffled flask (220 rpm shaking, 30 °C). Cells first reached confluency after 2 days of growth ($OD_{600}$ ~12), at which time 200 μL (which includes a sufficient number of cells for approximately 500-fold library coverage) was used to inoculate 25 mL of fresh media. The cells were again subcultured upon reaching confluency at day 4 for the growth screen, and the experiment was halted after 6 days of growth. At each time point (i.e., days 2, 4, and 6), 1 mL of culture was removed and treated with DNase I (New England Biolabs; 4 and 25 μL of DNaseI buffer) for 1 h at 30 °C to remove any extracellular DNA. Cells were isolated by centrifugation at $4500 \times g$ and the resulting cell pellets were stored at −80 °C for future analysis.

**Library isolation and sequencing**. Growth screen samples were thawed and resuspended in 400 μL sterile, milliQ $H_2O$. Each cell suspension was split into two, 200 μL samples, and plasmids from each sample were isolated using a Zymo Yeast Miniprep Kit (Zymo Research). Splitting into separate samples here was done to accommodate the capacity of the Yeast Miniprep Kit. The split samples from a single pellet were then pooled, and plasmid copy number was quantified using quantitative PCR with qPCR-GW-F and qPCR-GW-R and SsoAdvanced Universal SYBR Green Supermix (Biorad). Each pooled sample was confirmed to contain at least $10^7$ plasmids.

To prepare samples for next-generation sequencing, isolated plasmids were subjected to PCR using forward (ILU1-F, ILU2-F, ILU3-F, ILU4-F) and reverse primers (ILU(1-12)-R) containing all necessary barcodes and adapters for next-generation sequencing using the Illumina platform (Supplementary Table 10). Schematics of the amplicons from the Cas9 and Cas12a experiments submitted for NGS are pictured in Supplementary Fig. 7. At least 0.2 ng of plasmids (approximately $3 \times 10^7$ plasmid molecules) were used as templates, and PCR reactions were amplified for 16 cycles and not allowed to proceed to completion to avoid amplification bias. PCR product was purified using SPRI beads and tested on the bioanalyzer to ensure the correct length. Samples were pooled in equimolar amounts and submitted for sequencing on a NextSeq 500 at the UCR IIGB core facility.

**Generating sgRNA read counts from raw reads**. Next-generation sequencing reads were processed using the Galaxy platform[45]. First, read quality was assessed using FastQC v0.11.8. The reads were then demultiplexed using Cutadapt v1.16.6, trimmed using Trimmomatic v0.38, and mapped to each sgRNA using a combination of Bowtie 2 v2.4.2, and custom MATLAB scripts for counting bowtie alignments and naïve exact matching. Parameters used for each method are provided in Supplementary Table 11 and MATLAB scripts are provided as part of the GitHub link found below in the section "Data availability". Supplementary Table 12 provides further information correlating the NCBI SRA file names to the information needed for demultiplexing the readsets. Analysis of the CRISPR-Cas12a growth screens revealed that five sgRNAs were not present in the sequencing data. A pairwise comparison between normalized read abundances for biological replicates was done to verify consistency, see Supplementary Fig. 2 and Supplementary Table 1.

**CS analysis**. The CS associated with each guide was determined by taking the log2 of the ratio of normalized read counts of the control condition to the normalized read counts of the treatment condition. The control condition was taken as the normalized read counts at the end of the growth screen in a strain without Cas12a or Cas9. The treatment condition included constitutively expressed Cas9 or Cas12a with disrupted KU70. Normalized counts were taken as the total number of reads for a given sgRNA divided by the total reads for the corresponding sample. If no reads were identified for a given sgRNA, a pseudo-count of one was added to the read count to facilitate subsequent calculations. In all cases, normalized read counts

for each biological replicate were averaged together to produce an average normalized read count and associated standard deviation for each sgRNA. All normalized read counts and CS values are provided in Supplementary Data 3 and 4.

**Nucleosome occupancy analysis**. To account for genomic features, specifically nucleosome occupancy, we determined an average normalized occupancy score (ranging from 0 to 1) for every target locus using previously published MNase-Seq coverage data[46] (Supplementary Data 5). Per base nucleosome occupancy scores were summed up for each sgRNA, averaged, and normalized to a value between 0 and 1 by taking its ratio to the highest averaged value. This information was integrated into DeepGuide via a separate FCCN, the first step of which was to convert the one-dimensional occupancy data into an 80-dimensional real vector using a fully connected layer with 80 neurons. Using element-wise multiplication, the output of this layer was combined with the output of the last fully connected layer of the CS-predicting CNN to generate CS predictions that account for guide sequence, genomic context, and nucleosome occupancy.

**Validation of predicted sgRNA for Cas9 and Cas12a**. Four genes with easily screenable phenotypes, including MEF1, CAN1, MGA1, and RAS2 were selected for the validation of predicted sgRNA CS values (Supplementary Fig. 3). Gene sequences and the per base nucleosome occupancy of these genes were provided as input to the DeepGuide algorithm. As output DeepGuide predicted a CS value for each sgRNA of a given gene. sgRNAs were sorted from best to worst based on the predicted CS value from sequence-only (for Cas12a) and sequence plus nucleosome occupancy (for Cas9). The top 5 and bottom 5 sgRNA from the list were tested for editing efficiency.

To screen for RAS2 and MGA1 gene disruption, cultures with CRISPR plasmids growing in SD-Leu were diluted and plated in triplicate on YPD to obtain greater than 50 colonies on each plate. After two days of growth at 30 °C, the number of smooth colonies was counted and expressed as a fraction of the total colonies on the plate. For disruption of the CAN1 gene, cultures were similarly diluted and plated on YPD to obtain single colonies. Thirty colonies in triplicate were then randomly selected and streaked on SD-leu agar media supplemented with 50 mg $L^{-1}$ of L-canavanine. Colonies that grew on SD with canavanine were identified as positive for CAN1 disruption. To screen for MFE1, cultures were similarly plated, and 30 colonies from each transformation were randomly selected and streaked on SD-Oleic acid and dotted on YPD. Growth on YPD but not on SD-Oleic acid indicated MFE1 disruption. Screening of MFE1 was done on agar plates containing SD media supplemented with oleic acid as the sole carbon source (SD oleic acid; 0.67% Difco yeast nitrogen base without amino acids, 0.079% CSM (Sunrise Science, San Diego, CA), 2% agar 0.4% (v/v) Tween 20, and 0.3% (v/v) oleic acid).

**Reporting summary**. Further information on research design is available in the Nature Research Reporting Summary linked to this article.

## Data availability
The sgRNA sequencing data generated in this study have been deposited in the NCBI SRA database under accession code PRJNA766088. The sgRNA activity data (cutting scores) generated in this study are provided in the Supplementary Information/Source Data.

## Code availability
Source code for DeepGuide can be found at https://github.com/dDipankar/DeepGuide. Our GitHub page includes instructions for installation, usage examples. Custom MATLAB scripts that were used for the design of the Cas12a CRISPR library and processing the Illumina data to generate sgRNA abundance can also be found on the GitHub page. The Github repository has been archived to Zenodo to provide a permanent reference to the version of code used in this study [https://doi.org/10.5281/zenodo.5889577]. Generating sgRNA predictions for *Y. lipolytica* using DeepGuide does not require any specialized hardware and it can be carried out on a laptop with Conda installed.

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

## Acknowledgements

This work was supported by DOE DE-SC0019093 (to I.W. and S.L.), DOE Joint Genome Institute grant CSP-503076 (to I.W.), and NSF 1706545 (to I.W.).

## Author contributions

All authors conceived the idea and wrote the manuscript. A.R., C.S., and I.W. planned and analyzed the genome-wide CRISPR screens. A.R. conducted the CRISPR-Cas12a and guide-activity validation experiments. C.S. conducted the CRISPR-Cas9 screens. D.B. and S.L. planned the computational prediction of guide activity. D.B. designed and optimized the architecture of DeepGuide and collected data with DeepGuide and all other sgRNA prediction tools.

## Competing interests

The authors declare no competing interests.
