## [Peer Review File · Nature Communications]

Reviewers' Comments:

Reviewer #1:

Remarks to the Author:

This manuscript proposes an experimental-computational approach DeepGuide for sgRNA design specific to selected organisms. DeepGuide takes genome-wide data as input, uses unsupervised learning to obtain its representation, and then uses supervised learning to map sgRNA sequences, genome context, and epigenetic features with guiding activities. Subsequent calculations and experimental verification confirmed the effectiveness of this method in predicting activity sgRNAs. The whole manuscript clearly describes the methods and data sources, and the use of calculation and statistical data has been correctly implemented. I have some minor questions and comments:

- 1.The CNN model has been proposed for many years. Why did the author choose CNN instead of the newly proposed more advanced neural network algorithm?
- 2.The author has optimized the models of Cas9 and Cas12a respectively, and it can be seen from the "Ablation analysis" chapter that the difference of the model network structure will also affect the final prediction results. Will these factors affect the generality of the model?
- 3.The manuscript uses the 32-mers method for Cas12a. Is it also 32-mers for Cas9? Why the author chose 32-mers instead of other K values is not explained in the article.
- 4.The resolution of Figure S5 is too low, and the legend is too small, it is not easy for readers to see the details.
- 5.The performance of DeepGuide has not been verified in multiple benchmark data sets. If the verification on multiple data sets is supplemented, the results will be more convincing.
- 6.Some recently published articles apply neural networks and supervised learning to biological problems, such as 29878118, 33734296, 32960766 and 33027025 (PMID). The author can refer to them appropriately.

Reviewer #2:

Remarks to the Author:

In this study, Baisya and colleagues report on the establishment of a deep learning algorithm based on experimental and computational approaches to enable a high activity CRISPR guide RNA design (DeepGuide). This new tool focuses on the usage of the widespread SpCas9 and the propagating LbCas12a and includes not only "standard" organisms, but also non-conventional organisms such as fungal species and prokaryotes. The authors describe the existence of a variety of prediction tools designed for individual Cas proteins and/or individual model organisms, but they highlight the deficiency of an overarching tool that facilitates comparability and flexibility in CRISPR-Cas experiments. The authors successively guide through their genome wide CRISPR assay, the architecture of the DeepGuide algorithm, the optimization of the algorithm and its validation. The underlying bioinformatics of DeepGuide are elaborately described and provide a potentially significant contribution to gene editing technology by combining the deep learning technology with this broad application. However, in the context of a minor revision, Baisya and colleagues may address the following points to clarify some statements and to increase the strength of others.

1. --- The authors state DeepGuide to have the capacity to design highly active sgRNAs for both types of Cas proteins (SpCas9 and LbCas12a) in the context of a broad spectrum of species. The first point is addressed in a very diligent way by many comparisons and results. Whereas the proof of the applicability of DeepGuide for other species than *Y. lipolytica* is partially missing. Hence, a formal proof of this aspect by additional experimental work and/or meta-analyses of published experimentally validated sgRNA sequences for other species would increase the strength of the cross-species statement drastically.
2. --- In the first results and discussion paragraph, the authors state that no criteria have been used in order to generate the LbCas12a library. If the only criterion that has been applied was the proximity to the LbCas12a PAM sequence, most likely many sgRNAs with considerable off-target effects have been generated. How can be excluded that a high cutting score of a certain LbCas12a

sgRNA is not a false positive result due to off-target efficacy? Could this pose a potential vulnerability of the algorithm?

3. --- The authors diligently discuss the weakness of the SpCas9 dataset, which potentially hinders a better prediction power of DeepGuide due to the upfront bias towards highly active sgRNAs within the composition of the library. However, the LbCas12a part of DeepGuide is not powerful enough to give a "perfect" prediction either (which could of course only be expected in a utopian world). But in this context, the authors may add a paragraph discussing where there could still be potential for improvement and which factors may hinder such a "perfect" prediction.

4. --- The Cas9 protein is stated within the entire manuscript as "Cas9" with the exception of the figure legend title of Figure 1 and the figure legend text of Figure 2 in which it is stated as "Cas9a". Whether this is a typo or on purpose, for the sake of consistency this should be changed to a uniform term. In line with this, only the title of Figure 1 is in bold font, all others are not written in bold font.

5. --- Many sgRNA prediction tools for a variety of purposes are publicly available and easy to use. Hence, a new tool needs a better predictive power or a broader application or preferably both. In Figure 1b, the authors demonstrate an immensely enhanced correlation between the experimentally determined CS and the predicted CS of DeepGuide vs. picked out other tools. However, this superiority is based on the same dataset that was used to establish the algorithm. However, the same comparison with a published independent dataset of functionally validated sgRNA sequences would result in a real strong statement about the superiority of DeepGuide vs. other prediction tools.

6. --- In line with the previous point, it would be more clear, in order to judge the progress in the field of sgRNA activity prediction, if it would somehow be possible to quantify the prediction strength of DeepGuide in a numerical term such as for example x-fold better prediction precision as compared to the other tools using this independent dataset.

Best wishes

Reviewer #3:

Remarks to the Author:

The topic of this paper is primarily seen in bioinformatic prediction of Cas9 and Cas12a sgRNA guide activities. Especially the usefulness of DeepGuide, a machine-learning algorithm applied, is emphasized and claimed to provide better results than commonly used methods. This reviewer is not an expert in this topic and thus I will refrain from commenting on the impact of this part of the paper.

For *Yarrowia* the authors use one example in which a plasmid based sgRNA library is employed to induce double strand breaks and perform a screening of Cas proteins. The wetlab part is done well and successful but certainly does not warrant publication in NatCom alone as the methods are pretty standard and the outcome is not overly exciting. As I am not in a position to comment on the main part of the paper I can barely comment on the impact of the paper and its suitability for the journal.

Response to Reviewer #1

Expertise: ML/DL for predicting on/off target effects of genome editing proteins

This manuscript proposes an experimental-computational approach DeepGuide for sgRNA design specific to selected organisms. DeepGuide takes genome-wide data as input, uses unsupervised learning to obtain its representation, and then uses supervised learning to map sgRNA sequences, genome context, and epigenetic features with guiding activities. Subsequent calculations and experimental verification confirmed the effectiveness of this method in predicting activity sgRNAs. The whole manuscript clearly describes the methods and data sources, and the use of calculation and statistical data has been correctly implemented. I have some minor questions and comments:

1. The CNN model has been proposed for many years. Why did the author choose CNN instead of the newly proposed more advanced neural network algorithm?

Response: Thanks for this important question, which is often ignored in many studies that use deep learning to solve complex problems in molecular biology. As shown in the manuscript in Figure 4a and Figure 4b, we have selected for DeepGuide a CNN paired with the Convolutional Auto Encoder (CAE) only after carrying out a comprehensive analysis of many other well-established machine learning models, including random forests, support vector machines, logistic regression, gradient boosting regression, linear regression, and fully-connected neural networks. The chosen architecture clearly outperformed the others. We should note that many published methods for sgRNA prediction also employ CNNs. After the submission of the manuscript, we also tested recurrent neural networks (LSTMs), and tried to introduce an attention layer. Despite the additional complexity of these more sophisticated models (which can hamper interpretability and increase training time) we observed no significant improvement in the predictive performance. To emphasize the reasons behind our selection of a CNN approach, we have edited text in the discussion section of the manuscript.

2. The author has optimized the models of Cas9 and Cas12a respectively, and it can be seen from the "Ablation analysis" chapter that the difference of the model network structure will also affect the final prediction results. Will these factors affect the generality of the model?

Response: Thank you for raising this point. In our work, we have always tried to take into account the generalizability of our model by visualizing and comparing the training and validation loss function during our hyper-parameter and structure optimization. For instance, Figure S6 shows that the pre-training step (embedded in the weights of the Convolutional Auto Encoder) enhances the generalization abilities of DeepGuide (observe that the validation loss decreases with the number of epochs). Also shown in Figure S5 is that DeepGuide without pre-training archives a lower AUROC than DeepGuide with pre-training.

Having said that, it is clear from our work (as the reviewer correctly points out) and from a close reading of the published examples that take a deep learning approach to CRISPR guide design, that hyper-parameter and structure optimization will be necessary for each new data set; this is

expected. That is, each deep neural network method for CRISPR guide activity prediction is highly optimized for the dataset(s) it was trained and tested on, and ours is not an exception. In response to the point raised here as well as a similar point raised by Reviewer #2 (see comment #1), we have added discussion on the generalizability of our approach. This discussion highlights the fact that our overall approach is applicable to other species, but that high accuracy predictions are only possible through optimization on each new data set (see new Discussion section).

3. The manuscript uses the 32-mers method for Cas12a. Is it also 32-mers for Cas9? Why the author chose 32-mers instead of other K values is not explained in the article.

Response: We selected 32-mers for Cas12a and 28-mers for Cas9 based on the outcomes of a series of optimization experiments as shown in Figure 4. Our optimization experiments indicated that the shorter CRISPR guide length of Cas9 only required 28-mer for learning, while a longer, 32-mer, was optimal for Cas12a.

4. The resolution of Figure S5 is too low, and the legend is too small, it is not easy for readers to see the details.

Response: Thank you for bringing this to our attention; a higher resolution copy of Figure S5 is now included in the supporting information.

5. The performance of DeepGuide has not been verified in multiple benchmark data sets. If the verification on multiple data sets is supplemented, the results will be more convincing.

Response: We recognize that in our first submission we did not test the ability of DeepGuide to accurately predict guide activity from data sets generated in other organisms. In response to this comment and a similar comment from Reviewer #2, we tested DeepGuide on a published *E. coli* data set and a series of previously published data sets generated in mammalian cell lines. The results are now included as Figure S7. Briefly, with retraining DeepGuide was able to capture the activity of CRISPR guides in *E. coli* with equivalent accuracy as the gradient boosting trees method first used on this data set; Spearman, r values of both methods were equal to 0.542 (see ref. 35 and Figure S7). DeepGuide, however, was not able to predict guide activity in mammalian cells with similar accuracy ($r = \sim 0.33$ for DeepGuide vs. $r = \sim 0.85$ for DeepHF, the bidirectional-LSTM method first used on this data; see ref. 36 and Figure S7). Given the significant differences between the experimental measurements (*i.e.*, differences in how the guide activity data was generated) we did not expect that DeepGuide would accurately capture the mammalian cell data without re-optimizing the hyper-parameters (see response #2 above). Please also see our response to Reviewer #2, comment #1 for additional details on these new experiments.

6. Some recently published articles apply neural networks and supervised learning to biological problems, such as 29878118, 33734296, 32960766 and 33027025 (PMID). The author can refer to them appropriately.

Response: Thank you for the suggestions, we have now included additional references to support the use of deep learning in answering biological questions. The references added focus on CRISPR-Cas works as these are directly related to the submitted paper. These references include DOI:10.1038/s41467-019-12281-8 and DOI:10.1093/nar/gky572.

Response to Reviewer #2

Expertise: characterizing genome editing technologies, Cas9

In this study, Baisya and colleagues report on the establishment of a deep learning algorithm based on experimental and computational approaches to enable a high activity CRISPR guide RNA design (DeepGuide). This new tool focuses on the usage of the widespread SpCas9 and the propagating LbCas12a and includes not only “standard” organisms, but also non-conventional organisms such as fungal species and prokaryotes. The authors describe the existence of a variety of prediction tools designed for individual Cas proteins and/or individual model organisms, but they highlight the deficiency of an overarching tool that facilitates comparability and flexibility in CRISPR-Cas experiments. The authors successively guide through their genome wide CRISPR assay, the architecture of the DeepGuide algorithm, the optimization of the algorithm and its validation. The underlying bioinformatics of DeepGuide are elaborately described and provide a potentially significant contribution to gene editing technology by combining the deep learning technology with this broad application. However, in the context of a minor revision, Baisya and colleagues may address the following points to clarify some statements and to increase the strength of others.

1. --- The authors state DeepGuide to have the capacity to design highly active sgRNAs for both types of Cas proteins (SpCas9 and LbCas12a) in the context of a broad spectrum of species. The first point is addressed in a very diligent way by many comparisons and results. Whereas the proof of the applicability of DeepGuide for other species than *Y. lipolytica* is partially missing. Hence, a formal proof of this aspect by additional experimental work and/or meta-analyses of published experimentally validated sgRNA sequences for other species would increase the strength of the cross-species statement drastically.

Response: Thank you for your careful reading of our work. We agree, testing the ability of DeepGuide to predict CRISPR guide activity across different species and data sets will help demonstrate the generalizability of our method. Given that DeepGuide’s performance is optimal with a training set with upward of ~30,000 data points (see Figure 4), we sought to test the generalizability question on similarly sized data sets, including a ~70,000 data point set generated in *E. coli* (see Guo et al, ref. 35) and a series of data sets with at least 55,000 data points generated in mammalian cell cultures (see Wang et al., ref. 36). Notably, these data sets are for Cas9 and there are no publicly available datasets generated using Cas12a of this size.

We also note that the guide activity measurements made in mammalian cells were created using a synthetic library approach where target cut sites were integrated into the genome of the mammalian cells. The result of this approach (as acknowledged by the authors of the work, see ref. 36) is a bias towards genomically accessible cut sites, thus avoiding potential issues with guide accessibility due to chromatin structure. The *E. coli* data set is well-matched with the method we used to generate guide activity scores in *Yarrowia*; a Cas9 or Cas12a induced break in the native genome leads to cell death, which is used to determine guide activity. This method is a better reflection of CRISPR function as it accounts for genome structure and context.

The results of the cross-species comparison are now included as Supplementary Figure S7. DeepGuide was able to capture guide activity across the genome of *E. coli*; the Spearman value for DeepGuide, $r = 0.542$ is equivalent to that achieved in the original work by Gao et al. (see ref. 35; only Spearman values were reported). Re-training of DeepGuide on the *E. coli* data was necessary to achieve this value and in the absence of retraining predictions were poor (Spearman, r of 0.014). DeepGuide, however, was not able to accurately capture the mammalian cell data, achieving Pearson values of $r = 0.33$ or less on the three different data sets. Given the significant differences in how the data was generated, this lack of accurate mammalian cell guide activity predictions was expected. We anticipate that DeepGuide could be improved if optimized on the mammalian cell data sets.

One of the messages of our work is that no predictive method is truly species-independent, even if re-training takes place: one cannot expect very high predictive performance without an intensive optimization of the architecture for each data set as these data sets vary in terms of method used to acquire the data as well as a significant difference in genome structure between species. The new experiments described here (and now included in our manuscript) support this claim as retraining on new data sets improved the predictive power of DeepGuide, but did not produce the same performance as architectures that have been highly-optimized for a given species-specific data set. We have added discussion on this point to the manuscript as well as a description of these new results. In addition, we have edited the abstract to indicate that applicability to other species will require retraining as our new data suggests.

2. --- In the first results and discussion paragraph, the authors state that no criteria have been used in order to generate the LbCas12a library. If the only criterion that has been applied was the proximity to the LbCas12a PAM sequence, most likely many sgRNAs with considerable off-target effects have been generated. How can be excluded that a high cutting score of a certain LbCas12a sgRNA is not a false positive result due to off-target efficacy? Could this pose a potential vulnerability of the algorithm?

Response: Thank you for raising this point, we recognize that we were not sufficiently clear in this part of the main text. sgRNA in both the Cas9 and Cas12a libraries were designed with uniqueness in-mind, specifically, all sgRNAs passed a uniqueness check that is described in the 'sgRNA library design' section of the Methods. Our brief description of the library in the main text did not reflect this and has now been edited for clarity.

3. --- The authors diligently discuss the weakness of the SpCas9 dataset, which potentially hinders a better prediction power of DeepGuide due to the upfront bias towards highly active sgRNAs within the composition of the library. However, the LbCas12a part of DeepGuide is not powerful enough to give a “perfect” prediction either (which could of course only be expected in a utopian world). But in this context, the authors may add a paragraph discussing where there could still be potential for improvement and which factors may hinder such a “perfect” prediction.

Response: We agree that the manuscript would benefit from a short discussion on how one might improve predictions, both in terms of generating the data set and in predictive algorithms. This discussion can be found at the end of the manuscript.

4. --- The Cas9 protein is stated within the entire manuscript as “Cas9” with the exception of the figure legend title of Figure 1 and the figure legend text of Figure 2 in which it is stated as “Cas9a”. Whether this is a typo or on purpose, for the sake of consistency this should be changed to a uniform term. In line with this, only the title of Figure 1 is in bold font, all others are not written in bold font.

Response: Typos fixed, thank you.

5. --- Many sgRNA prediction tools for a variety of purposes are publicly available and easy to use. Hence, a new tool needs a better predictive power or a broader application or preferably both. In Figure 1b, the authors demonstrate an immensely enhanced correlation between the experimentally determined CS and the predicted CS of DeepGuide vs. picked out other tools. However, this superiority is based on the same dataset that was used to establish the algorithm. However, the same comparison with a published independent dataset of functionally validated sgRNA sequences would result in a real strong statement about the superiority of DeepGuide vs. other prediction tools.

Response: This is a good suggestion, thank you. In response to comment #1 above and to a similar concern raised by Reviewer #1, we have now evaluated DeepGuide's performance on other data sets, including *E. coli* and mammalian cell data sets that are similarly sized to the data we generated in *Yarrowia*. The new data shows that DeepGuide is able to capture *E. coli* CRISPR-Cas9 activity once retrained. We also note that we are not advocating that DeepGuide is a general all-species tool without retraining and/or significant optimization on new data sets. In fact, the current literature as well as our analysis in this manuscript suggests the opposite, that is, in order to obtain highly accurate CRISPR activity predictions, an optimized architecture along with a robust and precise training set is necessary. See response to comment #1 above for additional discussion.

6. --- In line with the previous point, it would be more clear, in order to judge the progress in the field of sgRNA activity prediction, if it would somehow be possible to quantify the prediction

strength of DeepGuide in a numerical term such as for example x-fold better prediction precision as compared to the other tools using this independent dataset.

Response: This is an interesting suggestion, but it would be difficult to develop a metric or an evaluation method that would replace the commonly used Spearman or Pearson coefficients. Since the output of DeepGuide is a predicted numerical score (and not a binary answer), the most natural way to measure the similarity between two ranked lists of scores (predicted vs. ground truth) is a correlation coefficient (*i.e.*, Pearson or Spearman). Spearman and Pearson coefficients are now *de facto* the metrics accepted by the community in sgRNA activity prediction. Every prediction tool published so far (including those we compared in the manuscript, namely SSC, sgRNA Scorer 2.0, CRISPRater, Designer v1 and v2, TSAM, CRISPRon, DeepCRISPR, and Seq-deepCpf1) uses Pearson or Spearman to measure the predictive performance, thus enabling comparison of the different methods across different studies.

Response to Reviewer #3

Expertise: Engineering *Yarrowia lipolytica* for industrial applications

The topic of this paper is primarily seen in bioinformatic prediction of Cas9 and Cas12a sgRNA guide activities. Especially the usefulness of DeepGuide, a machine-learning algorithm applied, is emphasized and claimed to provide better results than commonly used methods. This reviewer is not an expert in this topic and thus I will refrain from commenting on the impact of this part of the paper. For *Yarrowia* the authors use one example in which a plasmid based sgRNA library is employed to induce double strand breaks and perform a screening of Cas proteins. The wet lab part is done well and successfully but certainly does not warrant publication in NatCom alone as the methods are pretty standard and the outcome is not overly exciting. As I am not in a position to comment on the main part of the paper I can barely comment on the impact of the paper and its suitability for the journal.

Response: Thank you for agreeing to review our manuscript. We hope that you find our new CRISPR guide activity prediction tool useful in engineering *Yarrowia*. We note that our method of generating guide activity profiles across the genome produced some of the largest datasets of this kind. Moreover, we have not identified any other data set for Cas12a with tens of thousands of data points describing CRISPR activity in the native genome, suggesting that the dataset is a significant improvement over the current state-of-the-art.

Reviewers' Comments:

Reviewer #1:

Remarks to the Author:

The author addresses the issues raised relatively clearly. However, the author's references are only limited to CRISPR CAS related work, especially neural network related methods. In fact, machine learning can be applied to many different types of biological problems. Properly referring to the use of neural networks in other work can expand the author's ideas and track the latest computer technology. The authors are advised to make one more minor revision.

Reviewer #2:

Remarks to the Author:

All concerns have been addressed in the revision by Baisya, Wheeldon and colleagues.

Response to Reviewer #1

The author addresses the issues raised relatively clearly. However, the author's references are only limited to CRISPR CAS related work, especially neural network related methods. In fact, machine learning can be applied to many different types of biological problems. Properly referring to the use of neural networks in other work can expand the author's ideas and track the latest computer technology. The authors are advised to make one more minor revision.

Response: We agree with the reviewer and have added two references that point to the broader use of machine learning in understanding complex questions in biology. These references are provided in the concluding sentence of the manuscript.